# Clade-specific elemental signatures across an Early Triassic marine fauna pave the way for deciphering the affinities of unidentifiable fossils

Christopher P. A. Smith[1,2]*, Pierre Gueriau[3,4], Mathieu Thoury[4], Sebastian Schöder[5], Emmanuel Fara[2], Arnaud Brayard[2]

**1** CR2P, UMR 7207, CNRS, Sorbonne Université, Muséum national d'Histoire naturelle, Paris, France, **2** Biogéosciences, UMR 6282, CNRS, Université Bourgogne Europe, Dijon, France, **3** Institute of Earth Sciences, University of Lausanne, Lausanne, Switzerland, **4** Université Paris-Saclay, CNRS, ministère de la Culture, UVSQ, MNHN, IPANEMA, Saint-Aubin, France, **5** Synchrotron SOLEIL, L'Orme des merisiers, Gif-sur-Yvette, France

\* christopher.smith@mnhn.fr

## Abstract

In palaeontology, the observation of morphological characters is at the heart of species determination. Nonetheless, since most fossils have undergone considerable morphological loss, distortion, and/or flattening throughout their taphonomic history, the use of visual techniques often remains limited. Complementary approaches such as geochemical analyses or molecular palaeontology are increasingly developed. However, them as well remain limited by the preservation state and diagenetic overprinting of the vast majority of fossils. Based on data obtained by state-of-the-art non-destructive synchrotron micro-X-ray fluorescence (µXRF) major-to-trace elemental mapping of Early Triassic Paris Biota fossils, we show here, at least within a single fossil fauna, the existence of a clade-specific elemental signature. Using complete multi-elemental µXRF spectra instead of elemental quantifications/concentrations, we set a data-formatting protocol that allows us to compare the morphology of the spectra. We then statistically demonstrate the existence of a geochemical discrimination between specimens of different clade despite intra-clade mineralogical variability, and build a "elemental-comparative taxonomic identification" model accordingly. The latter, that goes beyond the simple distinction of tissue nature or type of preservation, is all the more important as it appears to hold the potential to identify some hitherto unrecognizable specimens of the fossil record.

## Introduction

Taxonomy in palaeontology has historically relied on visual observations of diagnostic anatomical characters under natural light. Such classical approach is evolving rapidly, particularly with the ongoing development and democratisation of 3D X-ray

**Data availability statement:** All relevant data are within the manuscript and its Supporting Information files available from Dryad : https://doi.org/10.5061/dryad.2rbnzs7v7.

**Funding:** This work was supported by the ANR project AFTER (ANR-13-JS06-0001-01), the French "Investissements d'Avenir" program (project ISITE-BFC: ANR-15- IDEX-03), the programme TelluS of the Institut National des Sciences de l'Univers, CNRS, and by the project e-COL+ "Valorisation des données naturalistes" managed by the Agence Nationale de la Recherche under the Programme d'Investissements d'Avenir with the reference ANR-21-ESRE-0053. Sample management and valorisation benefited from ANR-11-INBS-0004-RECOLNAT.

**Competing interests:** The authors have declared that no competing interests exist.

microtomography [1–3] and other advanced 2D imagery methods [4–8]. Additionally, new taxonomic tools have recently emerged such as the use of DNA and RNA extraction and sequencing from some rare, exceptionally well-preserved fossils up to 2 Ma old [9,10]. Another recent field of study for the taxonomy of extinct organisms is the investigation of indirect chemical traces/remnants (e.g., biomarkers) of previously existing organic material [11–14]. The taxonomic toolkit for palaeontologists is therefore quickly expanding. However, a major challenge remains, as most of the new and currently developed taxonomic tools focus on organic or organic-derived remnants and are therefore limited by the preservation state of the vast majority of fossils.

While earlier methods such as electron microprobe analysis and Particle Induced X-Ray Emission (PIXE) laid the foundation for non-destructive studies of geochemical signatures [15,16], these studies have recently taken a new step with the advances in analytic devices and the development of cutting-edge geochemical acquisition techniques such as synchrotron based micro X-ray fluorescence (μXRF) [17–19]. In particular, the latter has expanded analytical capabilities by allowing for the rapid acquisition of high-resolution elemental composition of fossils that may, once mapped, be used to reveal new anatomical structures and even discriminate different fossilized tissues [20–22]. Moreover, the precise quantification of particular elements has enabled the distinction between highly weathered tissues, even after the complete loss of the original material [23]. However, the use of these geochemical signatures remains, so far, essentially restricted to distinguishing different tissues [24,25], rather than taxonomic affinities. Besides, further applications of fossil geochemical signatures remain challenging due to the reliability and repeatability of elemental quantification. As it stands, the quantification of elemental compositions relies on the deconvolution of complex, often multi-elemental, spectra [26]. Although automatic deconvolution tools are regularly being developed [27–30], the selection of the present and absent elements remains user dependent, potentially introducing a strong user bias [30].

Here we present an innovative method that focuses on the morphology of the whole synchrotron μXRF multi-elemental spectra, rather than their derived quantitative elemental composition. Using this approach, we go beyond the distinction of some tissues or the mineralogical nature/type of preservation, and reveal a geochemical-based taxonomic discrimination among 249 million-years-old fossils sampled from the Early Triassic Paris Biota in western USA [31,32].

We first establish a data standardisation protocol that allows us to morphologically compare the spectra of the studied fossil specimens. We then apply statistical analyses to unveil the existence of a clade-specific geochemical signature among them, regardless of their locality of origin, the synchrotron beamline and associated setup used, or the large fossils intra-clade mineralogical variability. Following these analyses, we identify the zones of the spectra that are the most discriminant and only then, based on empirical data, do we interpret the geochemical signal. Finally, we propose a taxonomic identification model that may be applied to other, more or less well-preserved, undetermined specimens from the Paris Biota, and perhaps, in the future, to other key fossil material.

## Material

The Paris Biota [31,32] is a marine fossil biota presently documented in seven distant sites in the northern and western parts of the western USA basin (Fig 1). Two of these sites are located in northeastern Nevada, namely NoName and Immigrant Canyon (Fig 1C). Three are located in southeastern Idaho namely Paris Canyon, Stewart Canyon, and Georgetown (Fig 1D). We further add specimens from a new assemblage currently under study and located in northeastern Utah referred here to as LAK (Fig 1D; Brayard et al. ongoing work). The most distant sites are therefore located over 350 km apart. A seventh site, Montello Canyon, which neighbours Immigrant Canyon, is also regarded as associated with the Paris Biota [33].

The fossils from each site were collected from exposures of the Lower Triassic Thaynes Group *sensu* Lucas et al. [34]. The latter is essentially characterized by alternating limestones and shales of Smithian-Spathian age that correspond to relatively shallow, epicontinental marine depositional environments [35]. Based on ammonoid biostratigraphy [36–38], the Paris Biota is dated to the early Spathian, a time interval during which the western USA Basin was located at a near-equatorial latitude on the western margin of the Pangea (Fig 1B). More specifically, with the exception of the Immigrant Canyon site, all fossils were collected from early Spathian layers corresponding to the *Bajarunia-Tirolites-Albanites* beds [38]. Immigrant Canyon fossils are from slightly younger fossiliferous levels belonging to the *Prohungarites* and *Neopopanoceras* beds of middle-late Spathian age [32].

The Paris Biota is known for its exceptionally diverse, complex, and well-preserved fossils. It comprises a large variety of marine organisms such as algae, sponges [39], brachiopods, bivalves, echinoderms [33,40,41], cephalopods [38,42], vertebrates, coprolites [43], and arthropods [44–48]. This is all the more surprising as it is dated to less than 3 million

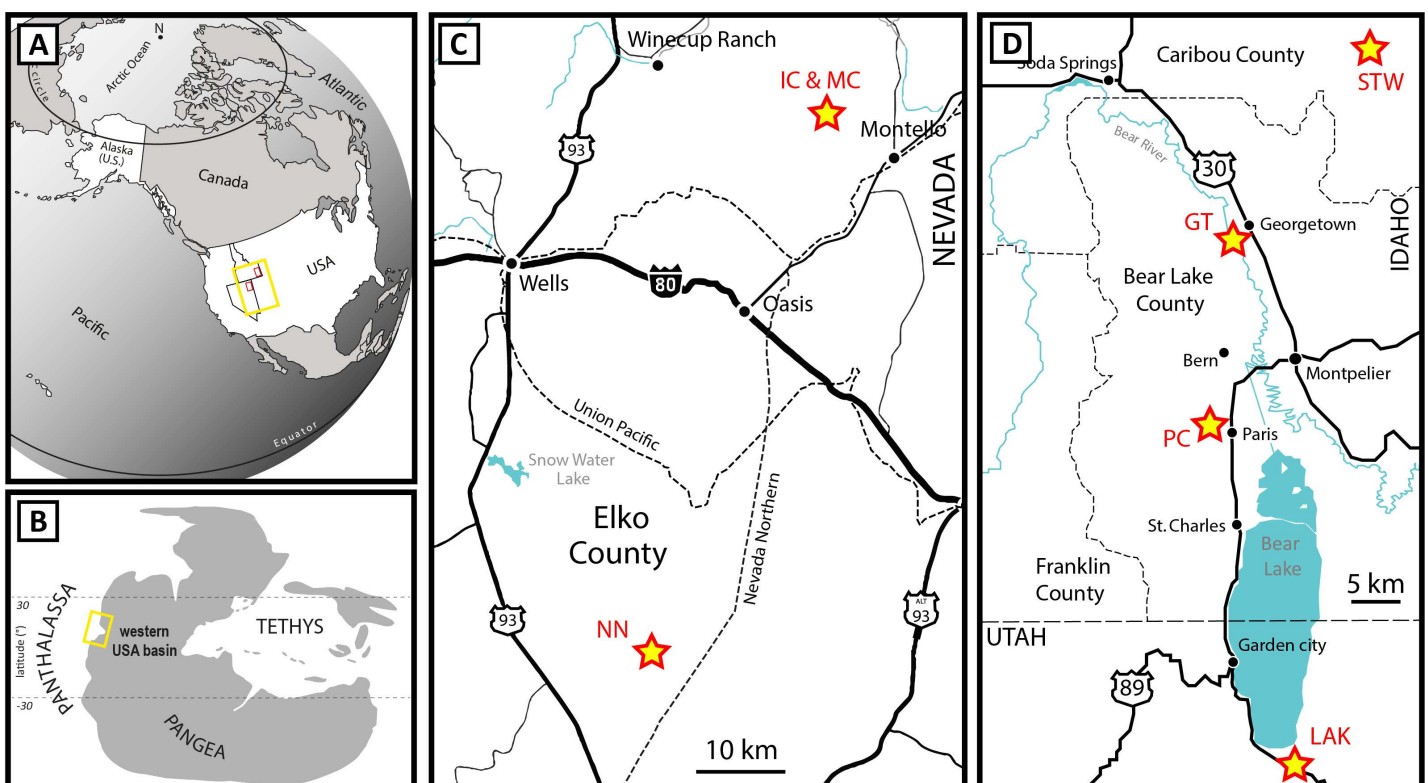

**Fig 1. Location of the sites from which the Paris Biota has been reported.** (A) Present-day map. (B) Early Triassic paleogeographic map. (C) Simplified map of northeastern Nevada. (D) Simplified map of southeastern Idaho and northeastern Utah. Based on Brayard et al. [31] and Smith et al. [32]. PC: Paris Canyon, GT: Georgetown, STW: Stewart Canyon, NN: NoName, IC: Immigrant Canyon, MC: Montello Canyon.

years after the Permian/Triassic mass extinction, i.e., the most severe biotic crisis of the Phanerozoic, which led to the extinction of over 80% of marine genera [49,50], and immediately after the Smithian/Spathian boundary (SSB; ~249.2 Ma) [51], which corresponds to a severe secondary extinction for nekto-pelagic organisms [52–55]. Together with the Guiyang Biota [56], the early occurrence of the Paris Biota challenges the commonly assumed paradigm of a sluggish and delayed post-Permian/Triassic biotic recovery [57–59].

In this study, we focus on 38 fossil specimens representing 8 high-level clades (ray-finned fish; ammonoids; gladius bearing coleoids; arthropods; echinoderms; brachiopods; sponges; lobe-finned fishes) from the seven known sites of the Paris Biota. The studied samples are listed along with their characteristics in Table S1 of the S2 File in S1 Text of Supporting information. From Paris Canyon, which is the best sampled site, we analysed by µXRF: one phosphatized fish; two phosphatized protomonaxonid sponges; one phosphatized gladius-bearing coleoid; three calcified ammonoids, two of which being poorly preserved (i.e., mostly molds with only very few remains of the mineralised fossil; poorly preserved ammonoids – PPA); two crinoids and one brittle star (echinoderms) preserved in calcite; four orbiculoid and one *Lingularia* brachiopod preserved in calcium phosphate; and four shrimps, two thylacocephalans, three stomatopods, and two lobsters (arthropods), all preserved as calcium phosphate. From LAK, we studied three shrimps (arthropods) preserved as calcium phosphate and one phosphatized putative gladius-bearing coleoid. From Stewart Canyon, we analyzed one ray-finned fish, one orbiculoid brachiopod, and one thylacocephalan (arthropod), all preserved as carbonaceous layers. From Immigrant Canyon, we studied one calcium phosphate protomonaxonid sponge and one brittle star (echinoderm) preserved in calcite. From Georgetown, we studied a phosphatized coelacanth (lobe-finned fish) scale, and we studied a carbonaceous shrimp (arthropod) from NoName. Only one crinoid (echinoderm), preserved in calcite, was studied from Montello Canyon. The mineralogical nature of each specimen was established based on observations and a previous work [60]. Although from different sites and of different mineralogical natures (e.g., phosphatic, calcitic, carbonaceous), all specimens were uncovered in compacted calcareous shales and are therefore preserved only in two dimensions.

## Methods

### Micro-X-ray fluorescence mapping

Micro-X-Ray Fluorescence mapping was performed at the SOLEIL synchrotron (France). Synchrotron-based µXRF is a non-destructive geochemical analysis technique whose principle is similar to that of energy dispersive X-ray spectroscopy (EDS) mapping, but whose incident source of irradiation is an extremely high flux of photons rather than electrons. Such a high flux of X-rays allows for the possible quantification of even very poorly concentrated elements and significantly decreases acquisition time, consequently increasing the maximum sample size [17]. Additionally, elemental information obtained by EDS is generally restricted to light elements at the sample surface and very close surface (a very few microns), whereas heavy elements can be probed, and to a certain depth (here ~80 µm) in µXRF, producing images in which the surface can be distinguished from the volume, revealing hidden anatomical features in some fossils [20].

The samples, which all comprise a fossil specimen embedded in sediment, were analysed on two synchrotron beamlines (PUMA and DiffAbs), through three experimental sessions (DiffAbs-2018, PUMA-2021 and PUMA-2024) with slightly different setups (Table 1), and from which the data were not exported the same way (e.g., different file formatting, different numbers of files, etc.). Additionally, the calibrated energy (X-axis of the spectra) was observed to slightly vary (20–25 eV) from one sample to another but well within the energy resolution of the detectors (>120 eV at Mn K$\alpha_1$). In order to homogenise the data, a post-acquisition reformatting (merging and structure standardization of the files) and realignment of each µXRF map was performed using R software (R Core Team 2019; Data S1 in S1 Text).

### Data processing

**Spectrum extraction.** The homogenised µXRF maps were processed using the PyMCA data-analysis software [29]. Firstly, the µXRF spectrum of each map was calibrated in energy based on the Ca (K$\alpha_1$ = 3.692 keV), Fe (K$\alpha_1$ = 6.405 keV),

**Table 1. Summary of the main specificities of the two setups used for data acquisition.**

| Synchrotron beamline | DiffAbs-2018 | PUMA-2021 | PUMA-2024 |
|---|---|---|---|
| Incident monochromatic X-ray beam energy | 18 keV | 18 keV | 18 keV |
| Beam size | 50 µm (diameter) | ~7 × 5 µm (Horizontal × Vertical) | ~7 × 5 µm (Horizontal × Vertical) |
| Data collection process | On the fly | On the fly | On the fly |
| Fly scan lateral resolution | 20-200 µm (pending on studied specimen) | 30-80 µm (pending on studied specimen) | 50 µm |
| Fly scan dwell time | 50 ms | 50 ms | 50 ms |
| Maximum surface coverage | 90 mm (both horizontally and vertically) | 150 mm horizontally, 100 mm vertically | 150 mm horizontally, 100 mm vertically |
| XRF Measuring device | Vortex-ME4, four-element silicon drift detector | SiriusSD silicon drift detector | SiriusSD silicon drift detector |

and Y (K$\alpha_1$ = 14.958 keV) peaks in order to obtain spectra of photon counts per energy, with energy ranging from ± 0.2 to ± 20.6 keV. Then, mean µXRF spectra of several seemingly homogeneous regions were extracted for each mapped sample (Fig 2A). In order to have pseudo-replicates, i.e., the same sample but not the same analysed area, a total of six mean µXRF spectra of selected regions were extracted per sample: three from the encasing sediment and three from the biomineralized fossil specimen itself. In total, we therefore obtained 252 spectra from the 38 studied specimens from both setups. To ensure consistency between sessions and to serve as control, two specimens were also analysed during all three experimental sessions.

**Spectrum formatting.** The µXRF spectra were exported as a succession of points rather than as a continuous curve (Fig 2B). The raw µXRF data (S1 File in S1 Text) are available in Supporting information. It also appeared that during data acquisition, the energy recording (X-axis of the spectra) slightly varied from one sample to another and, therefore, these points were not homologous between samples. Additionally, the data acquisition setups were optimised in order to detect elements ranging from Phosphorous, whose K-absorption edge (the minimum energy required to exit the K-shell electron) is 2.146 keV, to Uranium, whose $L_3$-absorption edge (the minimum energy required to exit the $L_3$-shell electron) is 17.166 keV. Therefore, the signal obtained outside this energy range is prone to a lot of noise and is thus unreliable. To overcome these issues, the continuous spectra were reconstructed using the exported data points (Fig 2C) and systematically subset from 1.7 keV to 18.3 keV (Fig 2D) using R software (R Core Team 2019). To simplify graphic reading, all spectra were then $\log_{10}$-normalized (Fig 2D). Finally, all spectra were scaled through standardisation, i.e., adjustment of the data so that it has a mean of 0 and a standard deviation of 1 (Fig 2D).

## Data analyses

**Geometric morphometric approach applied to the µXRF spectra.** To morphometrically characterize the µXRF spectra, landmarks were positioned at a regular interval along the X-axis (energy), every 0.02 keV (Fig 2D). Each spectrum is therefore morphologically described by 831 landmarks. In order to reduce the number of dimensions (in this case, the number of landmarks) while preserving the inter-spectrum shape variation information [61–63], we conducted a principal component analysis (PCA) on the 252 spectra.

Following this initial analysis ($PCA_0$), we observed a clear setup-specific spectra morphological differentiation, i.e., the data are distinct according to the experimental sessions and the associated setups (Fig 3A). Since two specimens had been analysed during each experimental session (DiffAbs-2018; PUMA-2021 and PUMA-2024), their µXRF spectra (2 specimens × 6 spectra × 3 setups = 36 spectra) were used to readjust the morphology of all spectra. First, the data acquired from PUMA in 2021 (PUMA-2021) was aligned with the data acquired from DiffAbs in 2018 (DiffAbs-2018; Fig 3B). To

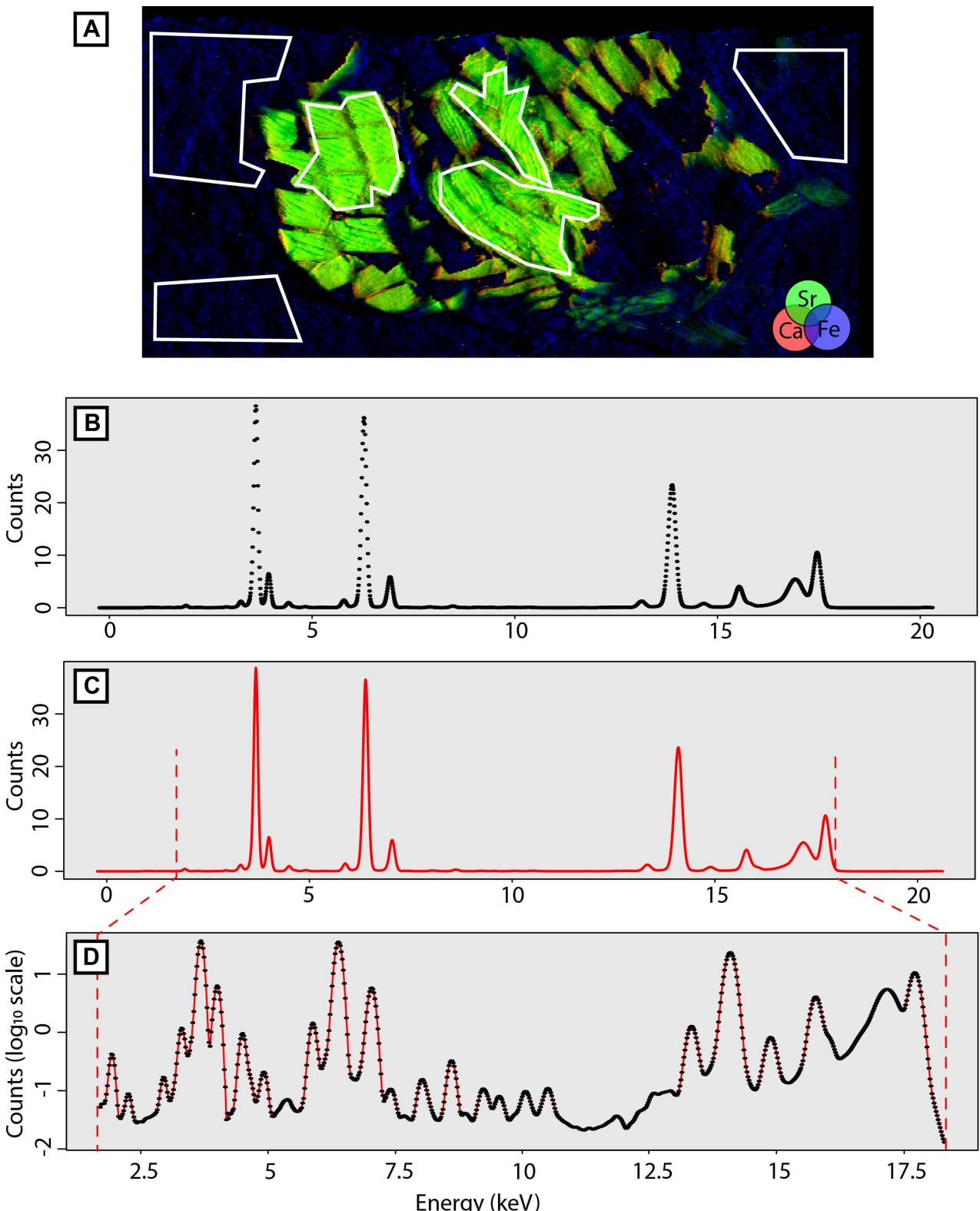

**Fig 2. Illustration of the successive steps of the data processing.** (A) Composite image of one of the analysed specimens (fish scales) in PyMCA. The white polygons delimit the zones from which the mean elemental spectra were extracted. (B) Raw data of a spectrum as exported from PyMCA. The

spectrum is described by a succession of discontinuous points at this stage. (C) Reconstruction of the continuous spectrum. (D) Final $Log_{10}$-normalized and standardized continuous spectrum from 1.7 to 18.3 keV. The points correspond to the landmarks extracted at a regular 0.02 keV step.

achieve this, the morphological difference between each pair of homologous spectra, i.e., spectra of the same region acquired during different experimental sessions, was calculated (Fig 3D). The mean of these morphological differences was then subtracted from all spectra acquired during PUMA-2021. The same approach (Fig 3E) was subsequently applied to adjust the data acquired from PUMA in 2024 (PUMA-2024) to align with the DiffAbs-2018 dataset, seemingly removing the vast majority, of the setup-specific morphological variations. Finally, to avoid any overrepresentation of the specimens analysed multiple times, their homologous realigned spectra from each experimental session were averaged in order to obtain a single spectrum per analysed zone. The PCA was then renewed ($PCA_1$) using the readjusted and averaged spectra (228 spectra; Fig 3C).

To investigate potential groupings of the studied spectra, cluster analyses on the coordinates of $PCA_1$ using three different methods, were first performed. The first was the *k*-means clustering method, which is based on minimising the total intragroup variance while optimising that of the intergroup [64,65]. The second method was hierarchical clustering, which is based on the similarity/dissimilarity between observations. The elements (observations and intermediary clusters) are clustered two by two depending on their distance (representing their similarity/dissimilarity), calculated here using the Ward method [66,67]. The resulting relationships are then presented as a hierarchical dendrogram. Finally, this dendrogram is "cut" at a specific height depending on the number of clusters sought. The third clustering method used was that of Gaussian Mixture Models (GMM) [68,69] that assumes that each observation is the result of a mixture of a specific number of Gaussian distributions with unknown parameters. The best-suited model was selected using the Bayesian Information Criterion (BIC) [70]. One of the main advantages of this clustering method is that it is a soft clustering method (i.e., the result of this clustering is a belonging probability of each observation to each cluster), unlike *k*-means and hierarchical clustering that are hard clustering methods (i.e., each observation is assigned to one or another cluster). Therefore, the accuracy of the clustering attribution of each observation can be evaluated.

Finally, linear discriminant analyses (LDA), also conducted on the coordinates of $PCA_1$, were carried out at different taxonomic resolutions, i.e., grouping the spectra of specimens under the same class depending on the taxonomic rank sought. Additional LDAs were conducted on another PCA in which the spectra of the coelacanth scale from Georgetown were removed ($PCA_3$), as this specimen was the only representative of his clade. The accuracy of each model was then tested by leave-one-out cross-validation, and using the contribution of each variable to the PCAs and LDAs, the most discriminant regions of the spectra were identified.

**Time series approach.** In addition to the morphometric approach, potential groupings were investigated using a method based on derived time series morphological descriptors. However, as this approach proved to be insufficient, only the geometric morphometric approach is presented in the main text. Details of the time series approach can be found in the S2 File in S1 Text available in Supporting information.

## Results

### Principal component analysis ($PCA_1$)

$PCA_1$ was conducted using the morphology of the μXRF spectra of all the studied specimens and the matrix in which they were embedded, once the setup-specific morphological differentiation was removed. The first four principal components of $PCA_1$ represent 79% of the whole μXRF spectra morphological variance (Fig 4A, D). A first observation is that once the data is re-adjusted using the specimens analysed on both setups, the setup-specific morphological differentiation is effectively eliminated (i.e., there is no clear distinction between the spectra obtained on different setups; Figs 3C and 4A). Additionally, although the studied specimens are from seven different sites, their broad overlapping across the first two

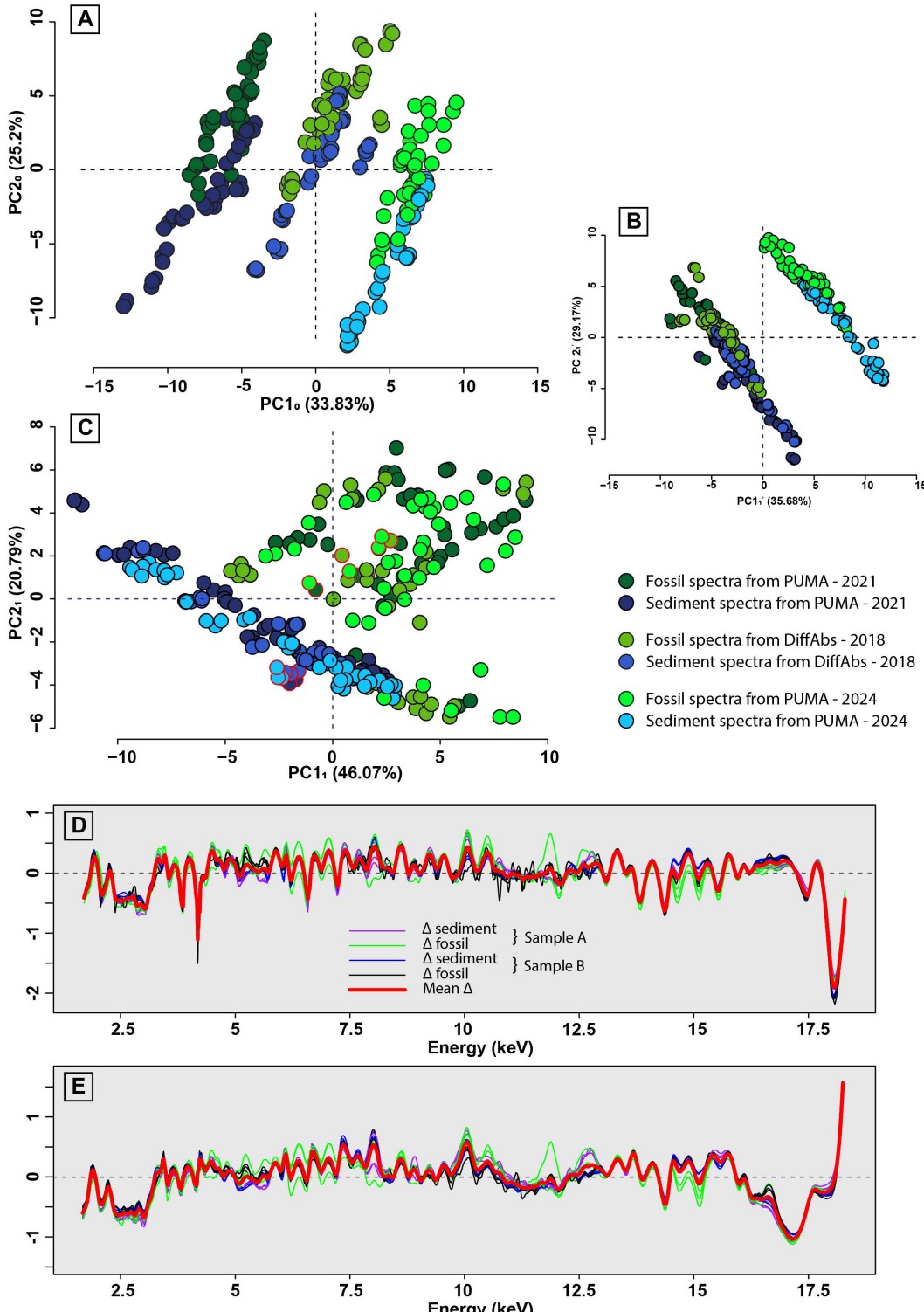

**Fig 3. Readjustment of the spectra acquired from the different experimental sessions (DiffAbs-2018, PUMA-2021 and PUMA-2024).** (A) Scatterplot along the two first principal components ($PC1_0$ & $PC2_0$) of $PCA_0$ showing a clear shift between the spectra acquired from each experimental session. (B) Scatterplot along the two first principal components ($PC1_{1'}$ & $PC2_{1'}$) of $PCA_{1'}$. i.e., PCA using the readjusted spectra of PUMA-2021 to DiffAbs-2018. The spectrum morphology at each pole of $PC1_0$, $PC2_0$, $PC1_{1'}$, and $PC2_{1'}$ may be found in Fig S1 of S2 File in S1 Text available in Supporting information.

(C) Scatterplot along the two first principal components ($PC1_1$ & $PC2_1$) of $PCA_1$, i.e., PCA using the readjusted spectra of PUMA-2021 and PUMA-2024 to DiffAbs-2018. The points circled in red correspond to the data used to readjust the spectra. (D) Morphological differences (Δ) between the homologous spectra acquired from PUMA-2021 and DiffAbs-2018. The mean Δ was used to readjust the spectra. (E) Morphological differences (Δ) between the homologous spectra acquired from PUMA-2024 and DiffAbs-2018. The mean Δ was used to readjust the spectra.

principal components suggest that site does not exert a dominant influence on μXRF spectra (Fig 4A). The positive pole of the first principal component ($PC1_1$, ~47% of the total variance) is characterised in particular by the high intensity of the peaks at 3.69 keV, 4.01 keV, 14.17 keV, 14.96 keV, and 15.84 keV (Fig 4B). On the other hand, the negative pole is characterised by the high intensity of the peaks at 3.31 keV, 4.51 keV, 4.95 keV, 5.9 keV, 6.41 keV, 7.06 keV, 8.05 keV, 8.64 keV, 9.25 keV, 9.57 keV, 10.54 keV, 12.65 keV, and 13.4 keV (Fig 4B). Additionally, there appears to be a slight intensity shift in the interval from ~1.7 keV to ~2.8 keV (Fig 4B), with peaks at ~1.95 and ~2.25 keV corresponding to Si escape peaks from Ca $K\alpha_1$ and $K\beta_1$ peaks at 3.69 and 4.01 keV respectively, higher in the positive pole. The main spectra morphological variations described by the second principal component ($PC2_1$, ~20% of the total variance) are the presence of peaks at 4.84 keV, 5.23 keV, 12.97 keV, and a strong peak at 14.96 keV towards the positive pole (Fig 4C).

The $PCA_1$ scatterplot with the first two principal components shows a rather clear distinction between the group "sediment + echinoderms + 2 PPA (poorly preserved ammonoids)" and the other fossil spectra (Fig 4A). The fossil spectra, excluding those of echinoderms and poorly preserved ammonoids, do not seem to present any clear taxonomic arrangement in the $PCA_1$ scatterplot. The "sediment + echinoderms + 2 PPA" spectra, on the other hand, exhibit a notable non-random distribution (Fig 4A). They are aligned, which highlights the existence of a morphological gradient (i.e., continuous evolution of one or more peaks along a gradient) within the "sediment + echinoderms + 2 PPA" spectra. Additionally, the echinoderm and ammonoid spectra are concentrated at one of the two extremities of this gradient, corresponding to the positive pole of $PC1_1$ and the negative pole of $PC2_1$.

To better describe this gradient, we computed a second principal component analysis ($PCA_2$) using only the sediment, echinoderm, and poorly preserved ammonoid spectra (Fig 5). The distribution of the observations in $PCA_2$ is congruent with the distribution of the observations in $PCA_1$ (Fig 5A). Additionally, the poles of the first principal component ($PC1_2$) in $PCA_2$ (Fig 5B) are congruent with the poles of the alignment observed in $PCA_1$. Therefore, $PC1_2$ represents well the morphological gradient existing within the "sediment + echinoderms + 2 PPA" spectra. The spectra morphological variation of $PC1_2$ is a combination of that of $PC1_1$ and $PC2_1$ (Figs 4B, C and 5B), which is consistent with the diagonal alignment of the "sediment + echinoderms + 2 PPA" spectra in $PCA_1$ (Fig 4A). Indeed, the positive pole of $PC1_2$ (~ 62% of the variance of $PCA_2$) is characterized by strong peaks at 3.69 keV, 4.01 keV, 14.17 keV, and 15.84 keV, and the absence of a peak at 14.96 keV (Fig 5B). This pattern could be associated to the echinoderm spectra and to remnants of an ammonoid signature conserved in the spectra of the two poorly preserved ammonoids, as they are concentrated at the positive pole of $PC1_2$ in the $PCA_2$ scatterplot. On the other hand, the negative pole is characterized by the presence of peaks at 6.41 keV, 7.06 keV, 8.64 keV, 9.25 keV, 9.57 keV, 10.54 keV, and 14.96 keV (Fig 5B).

## Clustering

Clustering analyses were performed using the six first principal components of $PCA_1$ accounting for ~85% of the total morphological variance of the data. Seeking firstly for a morphological discrimination between fossil and sediment spectra, a first set of clustering was constrained to two groups (Figs 6 and 7).

***k*-means clustering (Fig 6B).** When constrained to two groups, k-means clustering failed to separate fossils from sediments, likely due to the method's assumption of spherical clusters [71], which does not appear to hold in this dataset.

**Gaussian mixture model clustering (Fig 6C).** When constrained to only two clusters, the best-suited model was that of ellipsoids with equal volume and shape, and variable orientation (EEV). Based on this model, one of the clusters comprises only the sediment, echinoderm and 2 poorly preserved ammonite spectra, while the second cluster comprises

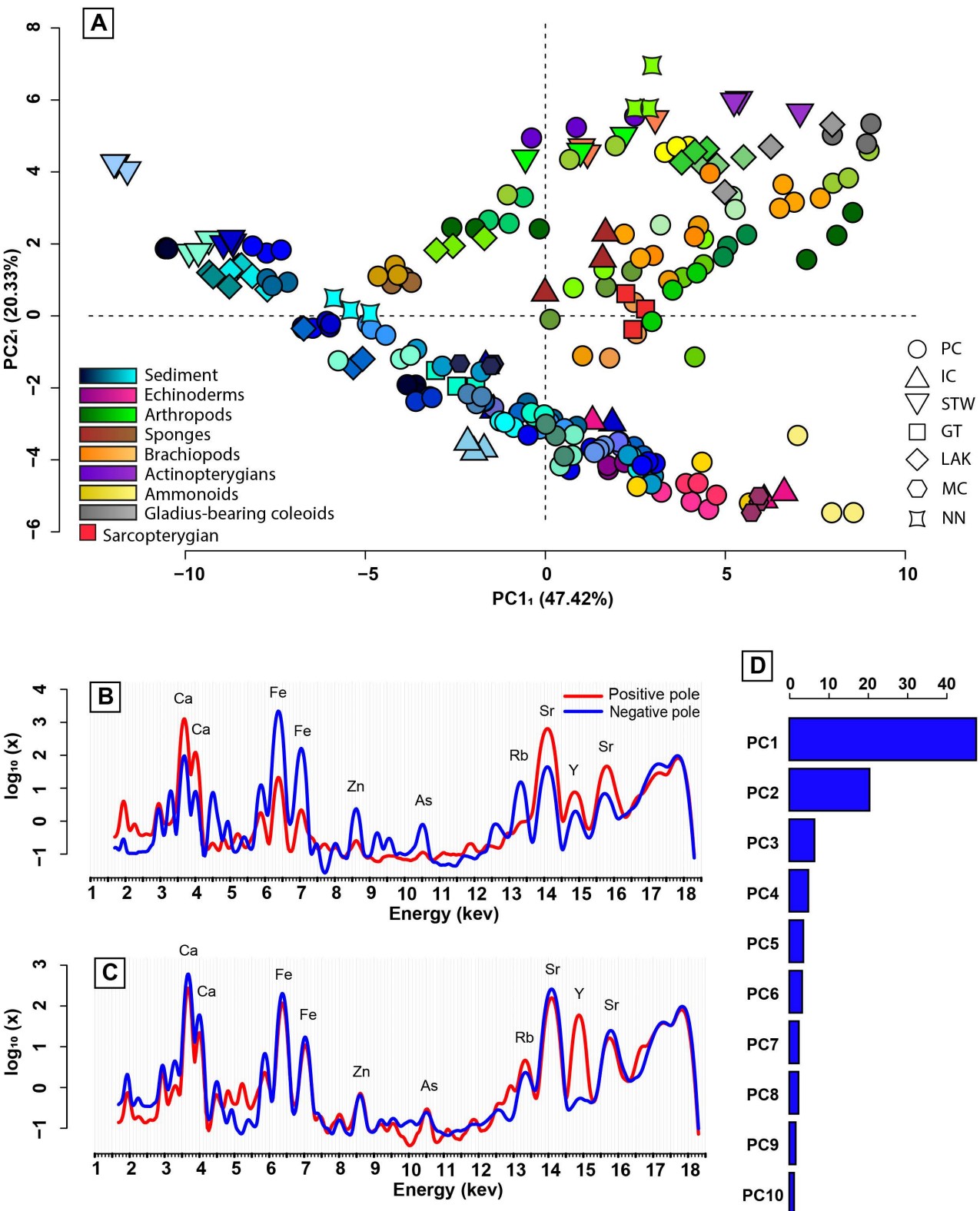

**Fig 4. PCA₁.** (A) Scatterplot along the two first principal components (PC1₁ & PC2₁). Data of the same colour represent pseudo-replicates of the same sample. (B) The morphology of the spectra represented at the positive and negative poles of PC1₁. (C) The morphology of the spectra represented at

the positive and negative poles of PC2₁. (D) The proportion of explained dispersion of the data for each principal component of PCA₁ in percentage. PC: Paris Canyon, GT: Georgetown, STW: Stewart Canyon, NN: NoName, IC: Immigrant Canyon, MC: Montello Canyon.

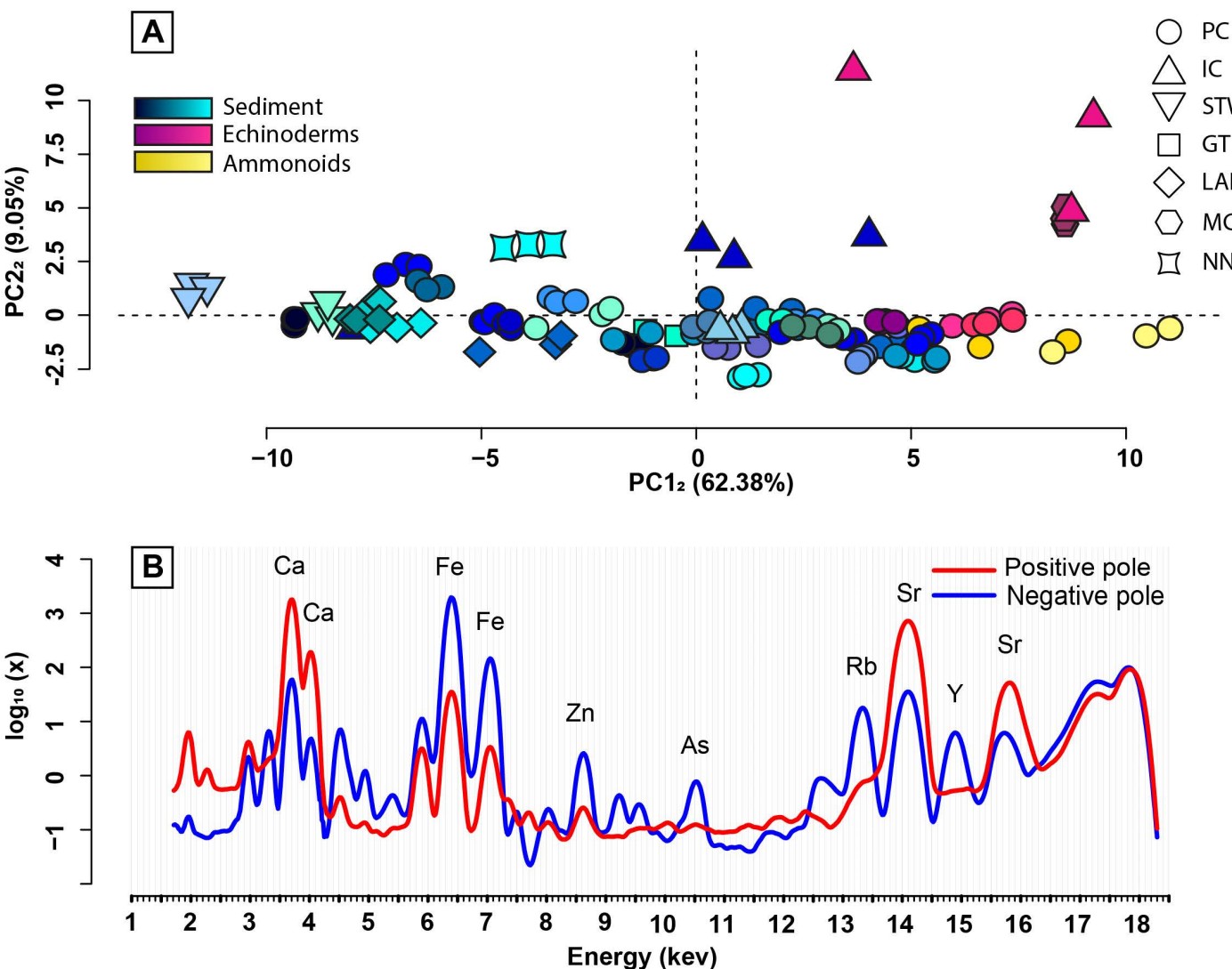

**Fig 5. PCA₂ i.e., PCA with only the sediment, echinoid, and poorly preserved ammonoid spectra.** (A) Scatterplot along the two first principal components (PC1₂ & PC2₂) of PCA₂. (B) The morphology of the spectra represented at the positive and negative pole of PC1₂.

all other fossil spectra. This empirically confirms the existence of a morphological difference between the spectra of the "sediment + echinoderms + 2 PPA" group and the spectra from the other fossils.

**Hierarchical clustering (Fig 7).** When cutting the hierarchical dendrogram built on the Ward distances of the PCA₁ scores to form two clusters, it appeared this clustering method was insufficient to fully discriminate the sediment spectra from the fossil spectra. Indeed, one cluster groups 75 sediment spectra alongside 6 sponge and 9 arthropod spectra, while the other cluster encompasses the rest of the fossil and sediment spectra. However, when the hierarchical dendrogram was cut to form three clusters, the previously mentioned cluster that grouped most fossil and sediment

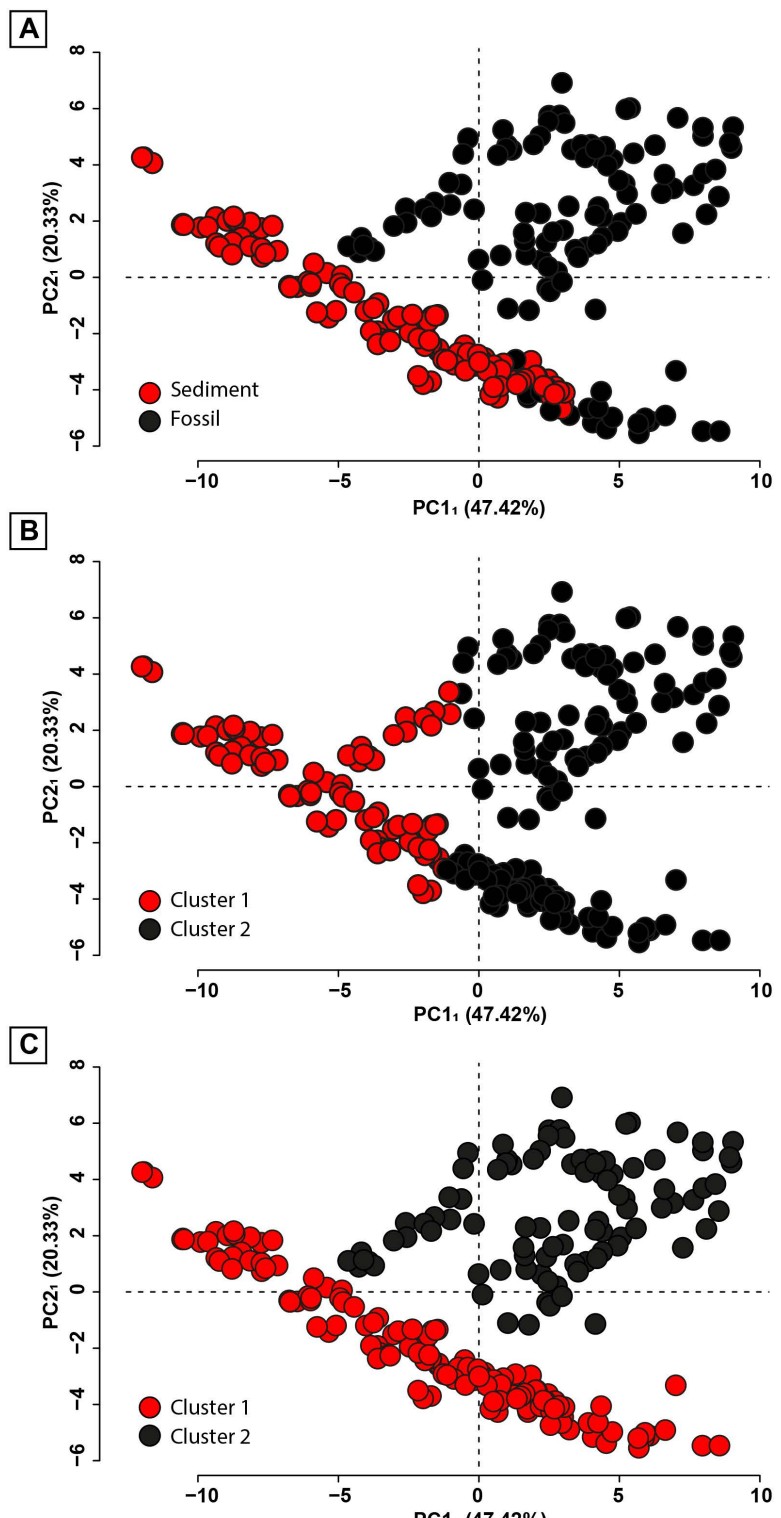

**Fig 6. Clustering analyses performed using the scores of the 6 first principal components of $PCA_1$.** (A) Scatterplot of the distribution of the sediment and the fossil spectra along the two first principal components ($PC1_1$ & $PC2_1$) of $PCA_1$. (B) *k*-means clustering for 2 groups. (C) GMM clustering for two groups.

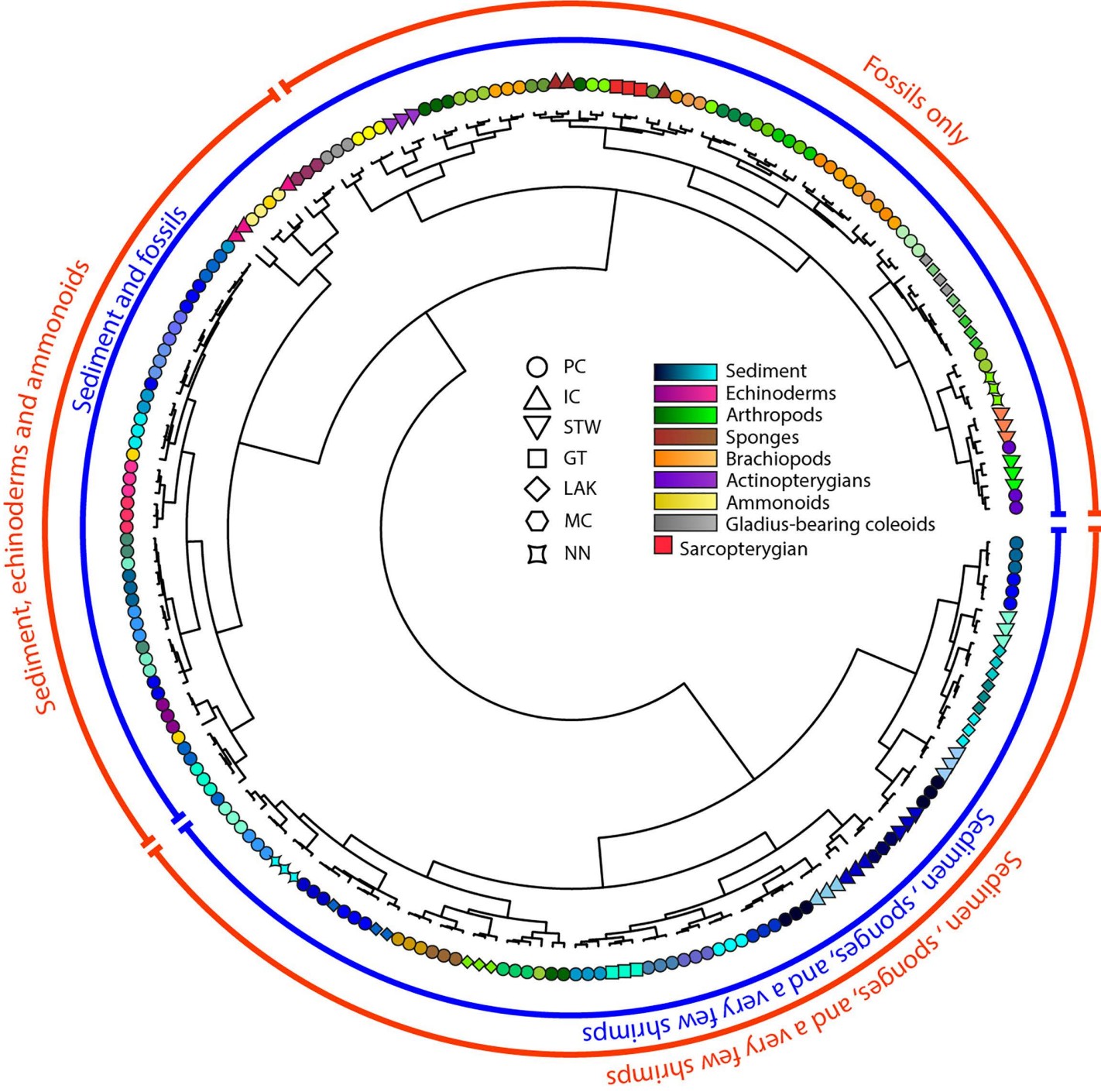

**Fig 7. Hierarchical clustering.** In blue for two groups; in red for three groups. The figurations are the same as in Fig 4: those of blue colour correspond to sediment spectra and those of other colours correspond to fossil spectra.

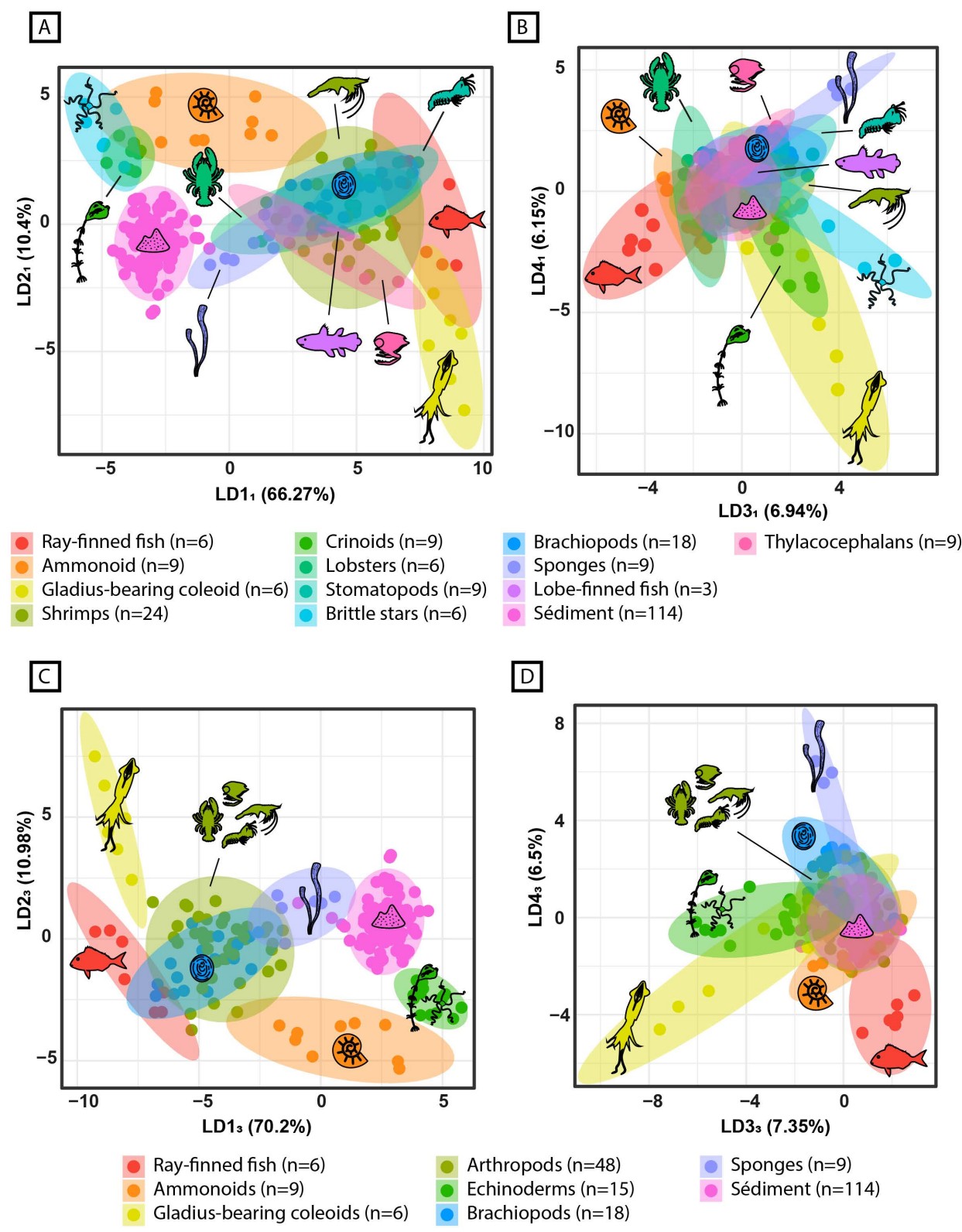

**Fig 8. Linear discriminant analysis at a moderately fine taxonomic resolution with all the data (LDA₁), and at a high-level clade resolution without singleton (LDA₃).** (A) LDA₁ scatterplot along the two first linear discriminant components. (B) LDA₁ scatterplot along the third and fourth linear discriminant components. (C) LDA₃ scatterplot along the two first linear discriminant components. (D) LDA₃ scatterplot along the third and fourth linear discriminant components.

spectra was split in two, and a fossil-only group was recognized. The latter comprises all fossil spectra with the exception of the previously mentioned sponge and arthropod spectra, the echinoderm spectra, and the spectra of two poorly preserved ammonoids.

   **Multi-group clustering.** The same clustering methods were applied to investigate additional potential groupings, including setup-specific, site-specific, and mineralogical categories. For the hard clustering analysis (*k*-means and hierarchical clustering), in which the number of clusters must be defined prior to the analysis, we computed multiple analyses with a gradually increasing number of clusters (from 4 to 15). However, none of these clustering analyses were able to correctly identify setup-specific, site-specific, or clade-specific clusters.

### Linear discriminant analysis on PCA scores

   **With all data on PCA$_1$ (Fig 8A, B).** A first linear discriminant analysis (LDA$_1$) was performed at a moderately fine taxonomic resolution (Class or Order; 13 groups) with all spectra (n = 228) using the scores of the first twenty-one principal components of PCA$_1$, accounting for over 98% of the whole morphological variance of the spectra (Fig 8A, B). After leave-one-out cross-validation, LDA$_1$ assigned the observation to the correct group in over 83% of the cases. A second linear discriminant analysis (LDA$_2$) was performed, again with all the spectra, and using the scores of the first twenty-one principal components of PCA$_1$, but this time at a high-level clade resolution (except for the ammonoids and the gladius-bearing coleoids that were kept distinct due to the very different original nature of their studied structures, i.e., internal gladius vs external shell), reducing the number of groups to 9 (Fig S2 of S2 File in S1 Text available in Supporting information). Leave-one-out cross-validation of LDA$_2$ revealed a ~89% correct identification rate.

   **Without singleton fossil (Fig 8C, D).** In order to avoid too much inter-group size disparity while also reducing the number of groups, the group represented by only one specimen (i.e., sarcopterygian spectra) was removed from the dataset. A principal component analysis was then performed again (PCA$_3$) on this amended dataset of 225 spectra, and another linear discriminant analysis (LDA$_3$) with a high-level clade resolution was performed using the scores of the twenty-one principal components of PCA$_3$ accounting for over 98% of the spectra morphological variance (Fig 8C, D). Leave-one-out cross-validation of LDA$_3$, which comprises 8 groups, revealed an ~89% correct identification rate, similar to that of LDA$_2$. However, singleton having been removed from LDA$_3$, the latter is statistically more robust than LDA$_2$.

   **Most discriminant regions of the spectra (Fig 9A, B).** Using the contribution of each variable (landmark) to each principal component in PCA$_3$, and the contribution of each principal component of PCA$_3$ to the construction of LDA$_3$, we identified the regions of the spectra that are the most discriminant. As expected, most of the discriminant regions are concomitant with the observed spectra morphological variance described by the two first principal components (PC1$_3$ & PC2$_3$) of PCA$_3$. However, a few other regions not initially identified in PC1$_3$ or PC2$_3$ seem to also contribute to the discrimination of the groups in LDA$_3$. These include the regions around 4.8 keV, 5.2 keV, 7.7 keV, and 13 keV. Additionally, other peaks, such as those around 5.4 keV, 8 keV, and 11.2 keV, appear to not be discriminant at all, while those around 7.5 keV, 11.9 keV, 12.65 keV, and 15.8 keV, are less discriminant than expected.

## Discussion

### Spectra (in)dependence

μXRF mapping is an increasingly exploited technique to quantify the elemental composition of various types of samples [72–75]. This is in particular due to the ongoing development of more accessible and precise analytic tools and setups over the last two decades [17–19]. In this study, two different synchrotron beamline setups were used through three experimental sessions. When comparing morphologically the μXRF spectra of the same samples acquired from during different experimental sessions, it appears there is a setup-specific dependence of the results, which is not surprising given their differences in design (from the insertion device to the detection system). Additionally, since the spectra

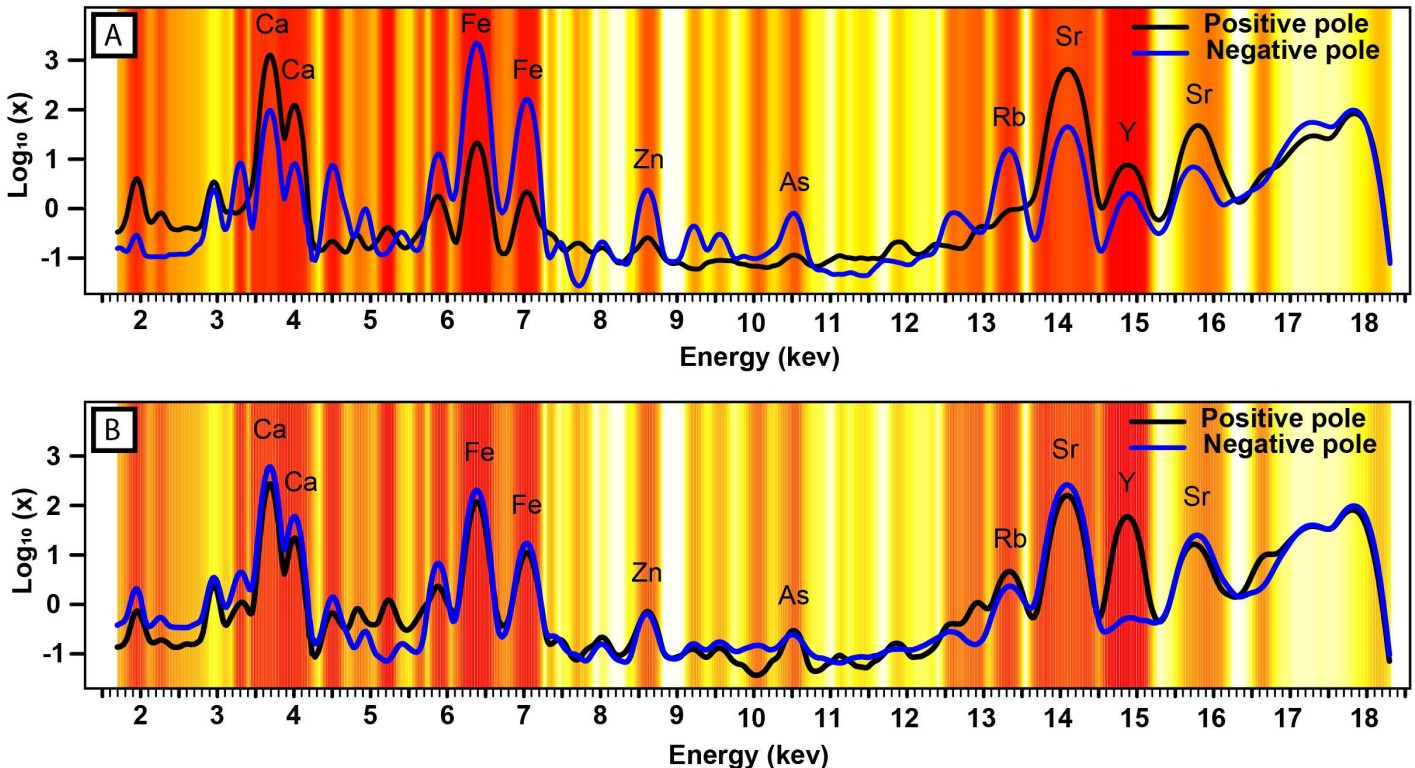

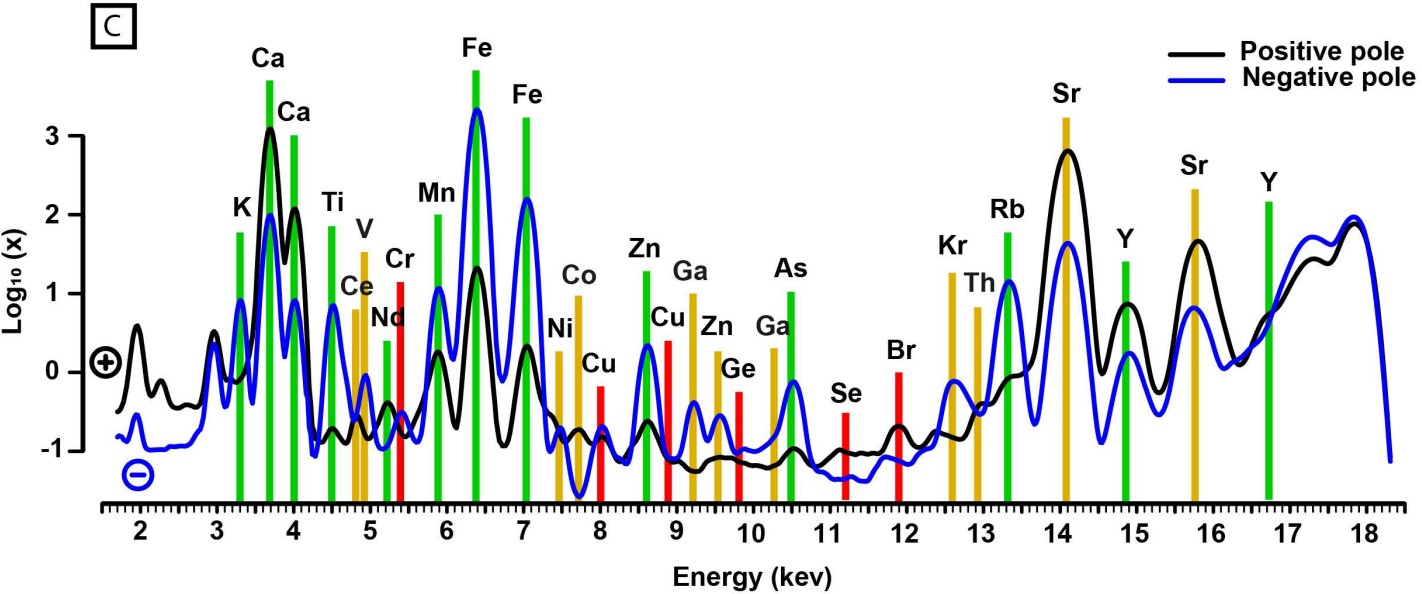

**High contribution to the discrimination**

**Moderate contribution to the discrimination**

**No contribution to the discrimination**

**Fig 9. Empirical interpretation of the geochemical signatures associated with LDA$_3$.** (A-B) Temperature diagram illustrating the regions of the spectra contributing the most (in red) to the discrimination in LDA$_3$. The curves correspond to the morphology of the spectra represented at the positive

and negative pole of: (A) the first principal component of $PCA_3$ ($PC1_3$); (B) the second principal component of $PCA_3$ ($PC2_3$). (C) Summary of the geochemical elements associated to the spectra morphological discrimination of $LDA_3$ model.

are $log_{10}$-normalized, this differential spectral morphology is of exponential amplitude. A significant proportion of the setup-specific spectra morphological differentiation occurs above 17.5 keV, where no elemental edges can be excited by the 18 keV beam, and only inelastic and elastic scattering processes, which are strongly dependent on the beamline setup, take place. However, although of lesser amplitude, setup-specific morphological differences also exist throughout the rest of the spectrum. This implies that absolute elemental quantification may not be fully comparable from one experiment to another.

Another approach is to focus on relative abundance rather than absolute elemental quantifications. However, our results show that the setup-specific spectra morphological differentiation is not constant throughout the spectra. Therefore, even relative abundance must be regarded with caution. Nonetheless, this setup-specific differentiation is quantifiable throughout the entire spectral range and can therefore be effectively corrected or minimized (see the "Geometric morphometric approach applied to the µXRF spectra" section in Methods).

Another variable is the origin of the samples. All the samples were collected from seemingly similar calcareous shales and are all attributed to the same biota. Yet, they come from seven different sites, up to 350 km apart from each other. Therefore, they may represent different depositional environments and may have been subject to different taphonomical processes. Neither the PCAs (Figs 4A and 5A) nor the clustering analyses (Figs 6 and 7) reveal any clear site-specific discrimination, not even among the sediment spectra. This evidences that, despite being from distant sites with potentially different geological/diagenetic histories, the sample spectra are comparable to each other, with limited site-specific bias.

Regarding the type of preservation (i.e., mineralogy) of the fossil specimens, only the spectra of the calcitic specimens appear grouped in the PCAs (Figs 4A and 5A), notably along a shared slope with sediments, potentially resulting from a similar carbonate composition. This is due to a shared carbonate composition. However, this observation is not confirmed by the clustering analyses (Figs 6 and 7). Additionally, all the calcitic specimens are echinoderms and poorly preserved ammonoids, implying this grouping may be of another nature, e.g., perhaps a taxonomical signal rather than (or co-varying with) a mineralogical signal. Therefore, apart from the possible exception of these two clades (see discussion in the following sections), there is no apparent mineralogy-specific morphological signature among the spectra of the fossils studied here, i.e., the morphology of fossil spectra is not dominated by mineralogical nature of the studied fossils. It should, however, be specified that these observations are constrained to the spectral range studied here (from 1.7 keV to 18.3 keV), and that it cannot be ruled out that a mineralogical signature exists through elements detectable at other (in particular, lower) energy ranges.

### Fossil spectra distinction (echinoderms and PPA excluded)

The most consistent clustering is observed using GMM on $PCA_1$ scores while seeking two groups (Fig 6C). It results in the differentiation between the "sediment + echinoderms + 2 PPA" spectra and the remaining fossil spectra. Even the spectra of sponges, which are particularly thin organisms, are correctly ascribed to the fossil-only spectra cluster. Considering the probed depth (~80µm), this feature indicates that the fossil signatures are sufficiently distinct from that of the sediment to be detected even if partially blended with the latter. The absence of pattern in $PCA_{features}$ (PCA conducted not on the full morphology of the spectra but using morphological features instead; S2 File in S1 Text available in Supporting information) and the low scores of $LDA_{features}$ (LDA carried out using $PCA_{features}$ scores) suggest, however, that the small loss of morphological information after extracting the spectrum features is sufficient to dissipate the discrimination. Therefore, to detect this discrimination, it is crucial to study the entirety of the spectra.

## Sediment, echinoderms, and poorly preserved ammonoids signature

The sediment, echinoderm, and poorly preserved ammonoid spectra appear aligned in the PCAs (Figs 4A and 5A), illustrating a morphological gradient well described by $PC1_2$ (Fig 5). The latter reflects an increase in calcium ($K\alpha_1 = 3.69$ keV; $K\beta_1 = 4.01$ keV) and strontium ($K\alpha_1 = 14.17$ keV; $K\beta_1 = 15.84$ keV) concentrations towards the positive pole of $PC1_2$ that is anticorrelated with an increase in iron concentration ($K\alpha_1 = 6.41$ keV; $K\beta_1 = 7.06$ keV) towards the negative pole of $PC1_2$ (Fig 2.B). Thus, the sediment varies from a rather calcitic composition to a more iron-rich one. Additionally, the increase in calcium concentration is accompanied by an enrichment in strontium ($K\alpha_1 = 14.17$ keV; $K\beta_1 = 15.84$ keV; Fig 5B). The latter most likely results from the former, as strontium is well-known for easily substituting itself with calcium [76,77] due to their identical oxidation number and similar atomic radii [78]. On the other hand, the increase in iron concentration is correlated with an enrichment in zinc ($K\alpha_1 = 8.64$ keV; $K\beta_1 = 9.57$ keV), gallium ($K\alpha_1 = 9.25$ keV), arsenic ($K\alpha_1 = 10.54$ keV), and rubidium ($K\alpha_1 = 13.37$ keV; $K\beta_1 = 14.96$ keV) (Fig 2B). The enrichment in gallium and arsenic likely originates from their substitution with ferric iron ($Fe^{3+}$), which is the most common form of iron [79], also due to their identical oxidation number and similar atomic radii [78]. Similarly, zinc and other transition metals likely substituted ferrous iron ($Fe^{2+}$).

The existence of this geochemical gradient within the sediments indicates that even within a same site and stratigraphic horizon, the geochemistry of the sediment is variable, underscoring the importance of collecting multiple sediment samples within a single locality and stratigraphic horizon when conducting geochemical studies. The origin of such variation, even within a single site that has undergone a specific history, is currently unclear, and additional analyses are required. Nonetheless, one hypothesis involves weathering processes that could have very locally (e.g., centimetres to metres) and differentially dissolved the calcium, leading to a relative increase in iron concentration. Another hypothesis is a potential differential concentration of organic material entrapped in the sediment, or a differential past microbial activity that may have locally induced a variable concentration of iron. While the sediment spectra of some sites may appear grouped, they remain within the wide morphological range of the sediment spectra of the best sampled site, i.e., Paris Canyon. Thus, the non-distinction of the sediment spectra from different sites also suggests that this spectral morphological gradient is congruent across all the sites studied here.

The echinoderm spectra are grouped in the "sediment + echinoderms + 2 PPA" cluster, at the extremity characterised by high concentrations of calcium and strontium (i.e., positive pole of $PC1_2$; Fig 5B). Additionally, in the PCAs they are distinct from the other fossil spectra (Fig 4). This strongly suggests that the echinoderm spectra are morphologically, and therefore geochemically, distinguishable from those of other fossils and from the sediment based on their calcium and strontium concentrations, which is congruent with the calcitic nature typical of this group [80].

The spectra of two poorly preserved ammonoids are also positioned at the extremity of the "sediment + echinoderms + 2 PPA" cluster, whereas the spectra of a well preserved ammonoid are confounded with the spectra of other fossils. This observation may be attributed to the fact that the spectra of the poorly preserved ammonoids represent a mixture of sediment and ammonoid signals. Their location, near echinoderm spectra in a region characterised by high concentrations of calcium and strontium, likely reflects the calcitic preservation nature of this group. However, as indicated by the position of the well-preserved ammonoid spectra, the complete ammonoid signal is more complex, distinguishing it from that of echinoderms.

## Taxonomical discrimination

After previously highlighting a morphological differentiation between the fossil and sediment spectra that extends beyond any potential site- or mineralogy-related effects, the remaining question is whether, or not, there is a taxonomic discrimination among the fossil spectra. The results of the $LDA_1$ (LDA at the class and order taxonomic level) confirm a morphological discrimination between the sediment and the fossil spectra. Furthermore, they support the existence of a taxonomic discrimination among the fossil spectra (Figs 8 and 9). However, $LDA_1$ was computed using thirteen categories

of very disparate sample sizes, with the smallest one (i.e., sarcopterygian) being represented only by one specimen (i.e., 3 spectra; Fig 8A, B). The poor ratio between the number of variables used to construct the LDA (twenty-one first principal components) and the number of categories implies that LDA$_1$ is prone to overfitting. Additionally, the statistical representativeness of some classes is highly questionable due to their small sample size. This is also the case for LDA$_2$ (i.e., LDA with all the data at the clade level; Fig 8C, D). However, when the specimens are grouped at the clade level and the singleton is removed, the number of classes is halved and the sample size of the smallest class is doubled. This significantly strengthens the statistical reliability of the analysis. The resulting LDA$_3$ again corroborates the existence of a taxonomic discrimination within the fossil spectra morphology (Fig 8C, D), showing a high score of leave-one-out cross-validation (~89%), validating the model. Such a cross-validation score is all the more significant as LDA$_3$ was built using specimens with variable preservation, and from different distant localities. This suggests that the LDA$_3$ model may be used to identify the high-level clade of a Paris Biota fossil specimen based solely on its whole µXRF spectra morphology, regardless of its origin or preservation type and quality.

## Geochemical interpretation

The taxonomic discrimination is based on the morphology of the spectra, which can be interpreted geochemically (Fig 9). As discussed previously, the PCA provides preliminary indications on the morphological and, therefore, the geochemical distinction of some of the groups. Using LDA$_3$, we refine this interpretation to distinguish the most morphologically discriminant zones of the spectra that are associated with specific chemical elements (Fig 9A, B). In this way, the calcium (K$\alpha_1$ = 3.69 keV; K$\beta_1$ = 4.01 keV) concentration does not only vary among the sediment, but also contributes significantly to the discrimination, most certainly as a marker of echinoderm and ammonoid fossils. Iron (K$\alpha_1$ = 6.41 keV; K$\beta_1$ = 7.06 keV) also appears as a discriminant element of the sediment, along with its likely associated gallium (K$\alpha_1$ = 9.25; keV, K$\beta_1$ = 10.27 keV, K$\beta_1$ = 16.74 keV) and arsenic (K$\alpha_1$ = 10.54 keV, K$\beta_1$ = 11.73 keV). Regarding yttrium (K$\alpha_1$ = 14.96 keV), it appears to contribute to the discrimination between the sediment and the fossils while being characteristic of the more iron-rich pole of the sediment, suggesting multiple yttrium concentration processes (substitution to Fe$^{2+}$ as well as to Ca in apatite). Other discriminant elements are potassium (K$\alpha_1$ = 3.31 keV), titanium (K$\alpha_1$ = 4.51 keV), manganese (K$\alpha_1$ = 5.9 keV), zinc (K$\alpha_1$ = 8.64 keV, K$\beta_1$ = 9.57 keV), and rubidium (K$\alpha_1$ = 13.4 keV). The peaks corresponding to these elements had already been identified in the PCAs. However, less evidently throughout the PCAs, cerium (L$\alpha_1$ = 4.84 keV), neodymium (L$\alpha_1$ = 5.23 keV), nickel (K$\alpha_1$ = 7.48 keV), cobalt (K$\beta_1$ = 7.65 keV), and thorium (L$\alpha_1$ = 12.97 keV) also appear to contribute to the taxonomic discrimination. Conversely, chromium (K$\alpha_1$ = 5.41 keV), copper (K$\alpha_1$ = 8.05 keV), germanium (K$\alpha_1$ = 9.89 keV), selenium (K$\alpha_1$ = 11.2 keV), and bromine (K$\alpha_1$ = 11.92 keV) do not seem to contribute to any discrimination. Additionally, vanadium (K$\alpha_1$ = 4.95 keV), gallium (K$\alpha_1$ = 9.25 keV; K$\beta_1$ = 10.27 keV), krypton (K$\alpha_1$ = 12.65 keV) and strontium (K$\alpha_1$ = 14.17 keV; K$\beta_1$ = 15.84 keV) only moderately contribute to the discrimination.

Evidently, understanding the how and the why of the differential incorporation of each of these elements, as well as their taphonomic implication, would require an element-specific thorough study. This is beyond the scope of this work, which focuses firstly on statistically identifying and characterising the differential and discriminant elemental composition of various fossil groups. Nonetheless, a few general hypotheses may be put forward to explain this potential clade-specific elemental discrimination:

(1) The original organic composition of the various clade – Indeed, not all clades produce the same biominerals [81–83]. Additionally, not all biominerals incorporate elements from the surrounding environment in the same way [84], and not all organisms have the same surrounding environment. Furthermore, initially different soft tissues may differentially influence the incorporation of elements into the biomineralized tissue [85]. The identified clade-specific elemental discrimination may be a remnant (i.e., a preserved geochemical signal despite various types of preservation) of the original differential elemental composition and/or the differential elemental incorporation between clade.

(2) The early taphonomic history – Recent developments in experimental taphonomy have confirmed that not all organisms undergo the same biostratinomic processes [86–89], and that different processes leave different geochemical signatures. Our clade-specific elemental discrimination may, therefore, actually reflect differential clade-specific biostratinomic processes, such as the influence of different post-mortem microbial communities, which may themselves be clade-specific.

(3) The diagenetic history – For instance, echinoderms are made of and generally preserved as calcite, whereas some other organisms may tend to undergo different recrystallization processes. There may be clade-specific preferential fossilisation pathways, however slight. This would indirectly lead to clade-dependent elemental compositions like those highlighted here.

### Geochemical fossil identification

Elemental mapping, whether through XRF analyses or other elemental mapping techniques, has already proven of great use to highlight, and sometimes even reveal, anatomical features barely visible to the naked eye [20,40,90–94]. Other multi-elemental studies have focused on discriminating tissues in fossil specimens [21,95]. Nonetheless, taxonomic identification in these studies remain based on the observation of anatomical diagnostic characters visualized through differences in relative elemental composition/abundance across a single fossil specimen. The approach proposed here, based on the morphology of the whole µXRF spectra of the Paris Biota specimens, shows the existence of a clade-specific signature that is common among the seven distant studied sites (i.e., no site-specific signal), and overcomes the different fossil preservation types (Fig 8C, D). This implies that the morphology of the spectra, and therefore the associated geochemical signature, may be used as a new taxonomic tool, free of user determination bias, to identify the high-level clade of the Paris Biota fossils, even and especially in the case of incomplete or poorly preserved remains unrecognisable to the naked eye.

As it stands, given the sampling used to build the models, and in particular the limited representativeness of some taxonomic groups, the most reliable model is that at the high-level clade resolution without singleton (LDA$_3$; Fig 8C, D). However, the results at a finer taxonomic resolution (LDA$_1$; Fig 8A, B) may suggest a discrimination even at the class and/or order level. At this time, the presented discriminant models and the associated spectra morphological discriminations remain restricted to the Paris Biota specimens and their associated palaeoenvironments. Nonetheless, if such discrimination is observable among the Paris Biota specimens, it is likely that similar models may be built and applied to other biotas, perhaps even from different geological contexts.

Furthermore, since the discrimination is based on the morphology of the whole spectrum, and not on the derived quantitative elemental composition of the specimens, other less precise spectra acquisition techniques (i.e., techniques with a usually unreliable absolute elemental quantification but that produce complete spectra, such as routine electron dispersive X-ray spectroscopy for instance) may be used to reveal similar spectral morphological differences. The spectra morphological discrimination is associated with specific elemental composition variations (Fig 9A, B). Here, we identified the most discriminant elements (Fig 9C). These elements may be used as references for seeking further geochemical discriminations among fossils, whether from the Paris Biota or from other fossil assemblages. Additionally, the elements highlighted in this study are those detectable within the studied energy range. Another focal area to explore would be to seek for similar patterns in the elemental composition of elements detectable at different energy ranges from those used in this study.

### Conclusion

Through the exhaustive morphological study of 252 synchrotron µXRF spectra of Paris Biota fossil specimens, and not solely by their interpolated geochemical quantification, we statistically demonstrate the existence of a clade-specific geochemical discrimination among fossils. Furthermore, the absence of clear locality and mineralogy distinction among

this discrimination suggest it is site- and preservation-independent. The resulting spectra morphology-based discriminant model (LDA$_3$) may therefore be used as a new taxonomic tool to identify the high-level clade of other Paris Biota fossils, regardless of their provenance, completeness, or mineralogical preservation. Preliminary results also suggest that this geochemical discrimination may be of higher taxonomical resolution than that of the high-level clade, encouraging further development of this approach on the Paris Biota and on other fossil assemblages.

This approach being solely based on the morphology of the spectra, it may also be applied to data acquired from other acquisition techniques, e.g., X-ray fluorescence data from lighter laboratory instruments such as benchtop µXRF or SEM-EDS, as long as the result is a complete spectrum. When doing so, however, one should pay close attention to potential setup-specific bias, as it may be significant. Furthermore, since this setup-specific bias is inconsistent throughout the spectra, elemental quantifications, whether absolute or relative, should be regarded with caution.

The most morphologically discriminant areas of the spectra were identified and may be geochemically interpreted. In this way, it appears that a high calcium concentration is a good marker of echinoderm fossils in the Paris Biota. On the other hand, iron, along with the associated gallium, and arsenic, appears as discriminant of the sediment, despite the latter exhibiting an elemental composition gradient from a rather calcitic to a more ferruginous pole. The other identified taxonomically discriminant elements among the Paris Biota specimens are potassium, titanium, neodymium, manganese, zinc, and rubidium. To a lesser extent cerium, vanadium, nickel, cobalt, gallium and thorium also contribute to the discrimination. Now identified, this set of elements may be used as a focal point for future taphonomical or coupled taxonomy-geochemistry studies.

Many questions remain to be solved, particularly regarding the how and the why of this taxon-specific differential elemental composition and its taphonomical implications. Nonetheless, the existence of a taxon-specific geochemical signature among fossils having now been revealed, it may henceforth be used as a new taxonomic tool, and perhaps even open the way to a new field of study that is "fossil comparative elemental taxonomy".

## Supporting information

**S1 Data.  S1 File.** This file contains: Data S1 - Pictures of the analysed samples exported from PyMCA with the studied zone highlighted by the shaded area. Data S2 - Mean µ-XRF spectrum of each studied zone. The data presents the mean value of photons measured per pixel ("counts" column) per energy level ("energy" column) over the sampled area. Data S3 - Mean µ-XRF spectrum of each studied zone regrouped per- specimens such as to be statistically analysed using R software **S2 File.** This document contains Figs S1, S2, Table S1, and supplementary text regarding the µXRF spectra morphological descriptors approach and the data acquisition setups. **S3 File.** This document is the R data analysis script. (DOCX)

## Acknowledgments

We thank E. Vennin, N. Goudemand, N. Olivier, G. Escarguel, J.F Jenks, K.G Bylund, D.A. Stephen and L.J. Krumenacker for their help during fieldworks. We thank M. McNamara for comments on an earlier version of the manuscript. We also thank the DiffAbs beamline staff.

## Author contributions

**Conceptualization:** Christopher Smith, Pierre Gueriau, Sebastian Schöder, Emmanuel Fara, Arnaud Brayard.

**Data curation:** Christopher Smith, Pierre Gueriau, Sebastian Schöder.

**Formal analysis:** Christopher Smith, Pierre Gueriau.

**Funding acquisition:** Mathieu Thoury, Sebastian Schöder, Emmanuel Fara, Arnaud Brayard.

**Investigation:** Christopher Smith, Pierre Gueriau.

**Methodology:** Christopher Smith, Pierre Gueriau, Sebastian Schöder.

**Project administration:** Christopher Smith, Mathieu Thoury, Emmanuel Fara, Arnaud Brayard.

**Resources:** Mathieu Thoury, Sebastian Schöder, Arnaud Brayard.

**Software:** Christopher Smith.

**Supervision:** Christopher Smith, Pierre Gueriau, Mathieu Thoury, Emmanuel Fara, Arnaud Brayard.

**Validation:** Christopher Smith, Pierre Gueriau, Sebastian Schöder, Emmanuel Fara, Arnaud Brayard.

**Visualization:** Christopher Smith.

**Writing – original draft:** Christopher Smith.

**Writing – review & editing:** Christopher Smith, Pierre Gueriau, Mathieu Thoury, Sebastian Schöder, Emmanuel Fara, Arnaud Brayard.

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
