## [Decision Letter · Decision Letter 0]

20 May 2025

Clade-specific elemental signatures across an Early Triassic marine fauna pave the way for deciphering the affinities of unidentifiable fossils

PLOS ONE

Dear Dr. Smith,

Thank you for submitting your manuscript to PLOS ONE. After careful consideration, we feel that it has merit but does not fully meet PLOS ONE’s publication criteria as it currently stands. Therefore, we invite you to submit a revised version of the manuscript that addresses the points raised during the review process.

       Thank you for your submission, and apologies for the delayed reviews. Please carefully consider the comments of both reviewers, both of whom along with myself, agree this is an interesting and applicable study, with some reservations. Reviewer 1 has included a word doc with specific comments and suggested edits. Reviewer 2's comments are listed below in the submitt4ed text. This is an interesting study demonstrating a potential utility to help resolve existing issues with fossils of unknown phylogenetic affinities.  I look forward to seeing your revisions and comments in your rebuttal letter should you choose to resubmit.  

We look forward to receiving your revised manuscript.

Kind regards,

David M. Lovelace, Ph.D.

Academic Editor

PLOS ONE

Journal Requirements:

2. In your manuscript, please provide additional information regarding the specimens used in your study. Ensure that you have reported human remain specimen numbers and complete repository information, including museum name and geographic location.

For more information on PLOS ONE's requirements for paleontology and archeology research, see https://journals.plos.org/plosone/s/submission-guidelines#loc-paleontology-and-archaeology-research .

6. We note that Figure 1 in your submission contain map/satellite images which may be copyrighted. All PLOS content is published under the Creative Commons Attribution License (CC BY 4.0), which means that the manuscript, images, and Supporting Information files will be freely available online, and any third party is permitted to access, download, copy, distribute, and use these materials in any way, even commercially, with proper attribution. For these reasons, we cannot publish previously copyrighted maps or satellite images created using proprietary data, such as Google software (Google Maps, Street View, and Earth). For more information, see our copyright guidelines: http://journals.plos.org/plosone/s/licenses-and-copyright.

Additional Editor Comments:

Thank you for your submission, and apologies for the delayed reviews. Please carefully consider the comments of both reviewers, both of whom agree this is an interesting and applicable study, with some reservations. There are several consistent themes between both reviewers that should be addressed. This is an interesting study with potential utility to help resolve existing issues with fossils of unknown phylogenetic affinities. I look forward to seeing your revisions and comments in your rebuttal letter should you choose to resubmit.

Reviewers' comments:

Reviewer's Responses to Questions

**Comments to the Author**

1. Is the manuscript technically sound, and do the data support the conclusions?

Reviewer #1: Partly

Reviewer #2: Yes

2. Has the statistical analysis been performed appropriately and rigorously?

Reviewer #1: Yes

Reviewer #2: Yes

3. Have the authors made all data underlying the findings in their manuscript fully available?

Reviewer #1: Yes

Reviewer #2: Yes

4. Is the manuscript presented in an intelligible fashion and written in standard English?

Reviewer #1: Yes

Reviewer #2: Yes

Reviewer #1: Smith and co-authors present an innovative exploratory investigation of paleontological applications of synchrotron micro-X-ray fluorescence utilizing a diverse set of metazoan body fossils and associated sediments. 

See attached file for review.

Reviewer #2: Here, Smith and coauthors present a suite of synchrotron beam x-ray fluorescence (XRF) data to explore elemental signatures of taxa across a suite of Paris Biota fossils from the Western USA. The authors use multivariate methods to compare full XRF spectra across the sample set. They show distinct differences in compositions at the clade level and some regional similarity in sediment/diagenetic composition. This work is interesting; however, it requires some revision to highlight the utility of this approach and to situate it better in the context of preexisting literature as it stands right now it may be overselling novelty and not highlighting the actual utility. I am recommending what may end up being some substantial revisions to the text, but support this work as being interesting and important to show both new analytical techniques and a useful statistical framework that circumvents some problems of the analytical technique.

1. Unclear how this is distinct from existing broad clade level distinction of biominerals? You already cite works like (Ulrich et al., 2021). There is no mention of ammonoids being originally aragonite? It seems that you are implying that debates like the affinity of Tullimonstrum can be solved with this clade-specific geochemical fingerprint, but at the phylum level this is too coarse to be useful, I think? This was the area that was most confusing to me because it seems like you are proposing a solution for a problem, but it is unclear how useful it can be in practice outside of a single formation where you have to calibrate with morphological identification then geochemical characterization and then finally unknowns could be classified.

2. There is an interesting note of possible diagenetic pathways, but this could use some expansion. For example in cephalopods show the soft tissue differences that cause dramatic differences in preservation (Clements et al., 2017).

3. Limiting this work to synchrotron x-ray sources means that it is not widely accessible. Is the goal application to small shelly fauna or Cambrian enigmas? Is it something that can be done with a handheld XRF or tabletop XRF or even EDS spectra given mapping across a surface? There is not a clear way to expand this work as it is currently presented to the readers.

4. I know it is beyond what you can do in this study, but to say you have a taxonomic discrimination tool that is independent of a priori taxonomic distinction without a broader dataset including modern specimens feels poorly supported. You can adjust by softening language probably or framing things as future research direction.

5. You state that site specific clusters do not exist, but I don’t think I see any visualization of the points coded by site after you standardize them. It would be useful to see this even to support the lack of site specific clustering.

I look forward to seeing these edits considered and the work improved. I believe that the multivariate approach to XRF spectra is needed to take full advantage of interactions without doing deconvolution techniques and the application of this to paleontology is interesting. Line by line and figure comments are contained below.

Line-by-line comments

Lines 55-59: Much of the capabilities of micro x-ray fluorescence recapitulates some of the capabilities of electron microprobe mapping across fossil materials? I agree prep is easier and throughput may be higher, but this seems to be less grounded in informing readers of the history of the field of elemental mapping across fossils

Lines 66-68: I think the deconvolution tools are likely and important thing to expand on here and in the discussion.

Lines 77-78: Mineralogical variability may be vague here. Variation in mineral implies calcite or aragonite or vaterite?

Line 123: Dryad data was a dead link when I tried to use it.

Line 125-139: Make this a table for quick comparison.

Line 126: Although the overall sample may have been a mold if you sample the shell with the XRF why the distinction between them?

Line 171-174: Was there a minimum area? What was the criteria for seemingly homogeneous? Why not just randomly subsample each area algorithmically and integrate that whole randomization procedure into what you’ve done? How does this step not add user-bias?

Line 200: Standardization meaning scaling peak to lowest count the same across spectra?

Line 206: The 831 landmarks presumably align with how many possible elements and their X-ray signature in your actual sample?

Lines 281-282: Why not just provide a most likely element for each peak?

Line 387: Internal gladius vs external shell are not mineralogical distinctions. Ammonites have shells made of the mineral aragonite and gladius composition is likely to also be aragonite, but may be comprised originally of chitin (Doguzhaeva and Mutvei, 2006).

Lines 466-467: I am confused about this statement. What does a mineralogy specific bias here mean? That aragonite or calcite or vaterite create secondary structure in the XRF spectra? Or do you mean that diagenetic alteration from aragonite to calcite does not drive any bias?

Lines 654 and 663: You use user-bias-free and setup-specific bias in these two sentences and I don’t quite follow the distinction. Is the user-bias an interpretive bias (which is still present given the need to explore clustering algorithms) or is it a setup-bias associated with instrumentation?

Figure comments

Figure 3. It would be useful if you include annotations for the element peaks associated with energy. Also, why do you not report or attempt to report the weights of the spectra on each PCA axis either with a simplification of using 1 or two energies per element to show the general discrimination? It would be a simple way to ground your discussion of the interpretation of the trends in multivariate space which you bring back to the elemental compositions anyway in your discussion.

Figure 4. Same comment on showing the elemental weights on the axes as in Figure 3. The sliding scale of color within types (e.g. sediment color gradient) is distracting, although I assume you did it to show distinctions between the points? Adjust alpha uniformly if overplotting is what you’re worried about. Each change in the visualization (e.g. x/y/color/symbol) should be data that can be interpreted.

Figure 9. Why are A and B in this replicate of C and D in Figure 8 with B being mirrored on the X axis? Why did the format of the x axis text in the spectra change so dramatically compared to the consistent constraints of the earlier figures? What do the background colors on the spectra mean in C and D and are they telling us anything other than what is visible with the positive or negative pole curve?

Figure 10. Again, why is the x axis formatted inconsistently (values, energy Kev) compared to earlier figures?

Work cited

Clements, T., Colleary, C., De Baets, K., and Vinther, J., 2017, Buoyancy mechanisms limit preservation of coleoid cephalopod soft tissues in Mesozoic Lagerstätten: Palaeontology, v. 60, p. 1–14, doi:10.1111/pala.12267.

Doguzhaeva, L.A., and Mutvei, H., 2006, Ultrastructural and chemical comparison between gladii in living coleoids and Aptian coleoids from Central Russia:, https://repository.geologyscience.ru/handle/123456789/45940 (accessed May 2025).

Ulrich, R.N. et al., 2021, Patterns of Element Incorporation in Calcium Carbonate Biominerals Recapitulate Phylogeny for a Diverse Range of Marine Calcifiers: Frontiers in Earth Science, v. 9, doi:10.3389/feart.2021.641760.

**Do you want your identity to be public for this peer review?** For information about this choice, including consent withdrawal, please see our Privacy Policy

Reviewer #1: No

Reviewer #2: No

---

## [Author Response · Author response to Decision Letter 1]

24 Jun 2025

We sincerely thank the editor for his editorial work, and both reviewers for their thorough, in-depth, and particularly constructive reviews. We are especially grateful for the high quality of the feedback, as we understand that this manuscript spans several fields (statistics, geochemistry, palaeontology), which may not be everyone's area of expertise.

We have carefully considered all the comments and suggestions made by the editor and reviewers. In addition, we made some minor modifications to the statistical analyses to enhance the rigor of our approach. Importantly, these changes did not alter the overall results.

Please find our responses to each reviewers comment in the response to reviewers document (see also tracked changes in the revised manuscript).

---

## [Editor Report · Decision Letter 1]

17 Jul 2025

Clade-specific elemental signatures across an Early Triassic marine fauna pave the way for deciphering the affinities of unidentifiable fossils

PONE-D-25-05930R1

Dear Dr. Smith,

We’re pleased to inform you that your manuscript has been judged scientifically suitable for publication and will be formally accepted for publication once it meets all outstanding technical requirements.

Kind regards,

David M. Lovelace, Ph.D.

Academic Editor

PLOS ONE

Additional Editor Comments (optional):

Thank you for submitting your edits, and very well documented changes. I believe this is suitable for acceptance at this point, and I appreciate the novel approach. My apologies for not getting this back to you sooner, I was in the field and unable to access the files to review appropriately. Well done, and than you for your patience!